



# Improving model-satellite comparisons of sea ice melt onset with a satellite simulator

Abigail Smith [1, 2], Alexandra Jahn [1], Clara Burgard [3, 4], and Dirk Notz [3, 5]

[1]Department of Atmospheric and Oceanic Sciences and Institute of Arctic and Alpine Research, University of Colorado Boulder, USA
[2]now at National Center for Atmospheric Research (NCAR), Boulder, USA
[3]Max Planck Institute for Meteorology, Hamburg, Germany
[4]now at Univ. Grenoble Alpes, CNRS, IRD, Grenoble INP, IGE, 38000 Grenoble, France
[5]Center for Earth System Research and Sustainability (CEN), University of Hamburg, Germany

**Correspondence:** Abigail Smith (abigail.l.smith@colorado.edu)

**Abstract.** Seasonal transitions in Arctic sea ice, such as the melt onset, have been found to be useful metrics for evaluating sea ice in climate models against observations. However, comparisons of melt onset dates between climate models and satellite observations are indirect. Satellite data products of melt onset rely on observed brightness temperatures, while climate models do not currently simulate brightness temperatures, and therefore must define melt onset with other modeled variables. Here

we adapt a passive microwave sea ice satellite simulator (ARC3O) to produce simulated brightness temperatures that can be used to diagnose the timing of the earliest snowmelt in climate models, as we show here using CESM2 ocean-ice hindcasts. By producing simulated brightness temperatures and earliest snowmelt estimation dates using CESM2 and ARC3O, we facilitate new and previously impossible comparisons between the model and satellite observations by removing the uncertainty that arises due to definition differences. Direct comparisons between the model and satellite data allow us to identify an early

bias across large areas of the Arctic at the beginning of the CESM2 ocean-ice hindcast melt season, as well as improve our understanding of the physical processes underlying seasonal changes in brightness temperatures. In particular, the ARC3O allows us to show that satellite algorithm-based melt onset dates likely occur after significant snowmelt has already taken place.

## 1    Introduction

Global climate models are important tools for understanding how Arctic sea ice is changing today and will change in the future. However, climate models show a large spread in their projections of Arctic sea ice area, and the causes of this spread are not well known (SIMIP-Community, 2020). In order to assess the fidelity of model simulations, and ultimately improve climate model representations of sea ice, we must evaluate model simulations against observations. Satellite observations of Arctic sea ice have pan-Arctic spatial coverage over four decades, and are thus well-suited for climate model evaluation. However, the

most common use of satellite observations for model evaluation in terms of sea ice area or extent provide a limited perspective on model differences, as they do not provide insights into why model differences exist. To do that, more process-oriented



metrics are needed (Notz et al., 2016). In fact, differences in the simulation of seasonal sea ice growth and melt have been shown to contribute to intermodel spread (Massonnet et al., 2018). Melt onset in particular was shown to be a useful metric for model assessment of thermodynamic processes, as models with biases in the timing of melt onset can produce realistic September sea ice areas for the wrong reasons (Smith et al., 2020).

Uncertainty as to which physical processes are captured by remote-sensing melt onset products, however, continues to complicate the use of these products as process-based metrics for model assessment (Jahn et al., 2012; Smith and Jahn, 2019; Smith et al., 2020). This is due to the fundamental difference in how melt onset, and other sea ice properties, are obtained in models and from remote sensing. In models, the physical evolution of the sea ice cover is simulated based on energy and momentum fluxes, including when snow on sea ice and sea ice itself begin to melt. In contrast, remote sensing products of melt onset are based on brightness temperatures from satellite retrievals, using an algorithm designed to detect the increase in emissivity that occurs when liquid water develops around snow grains (Smith, 1998; Drobot and Anderson, 2001; Markus et al., 2009; Kern et al., 2016; Bliss et al., 2017). However, there is uncertainty in which processes or thresholds these brightness temperature-based algorithms capture, as different algorithms (i.e., the advanced horizontal range algorithm (AHRA) (Drobot and Anderson, 2001) and the passive microwave (PMW) algorithm (Markus et al., 2009)) show large differences (Bliss et al., 2017). This uncertainty surrounding the physical conditions captured by remote sensing melt onset products causes a fundamental challenge for comparisons with models, as there are multiple possible definitions for sea ice melt onset in climate models. Modeled melt onset dates derived from different definitions can differ from each other and hence complicate the definite detection of a model bias (Smith and Jahn, 2019).

To reduce the uncertainty in the comparison of melt onset between models and satellites, we here use simulated brightness temperatures and a new, consistent metric for the earliest snowmelt that can be applied to both simulated and retrieved brightness temperatures. This approach allows for a more direct comparison between melt onset in models and satellite data than has been previously possible. By utilizing the earliest snowmelt metric, we can more precisely detect model biases using the satellite data, as well as provide insights into the physical processes captured by other melt onset definitions. Since the goal of this study is to perform the most direct model comparisons with satellite retrievals to establish the usefulness of the earliest snowmelt metric, we use an ocean-sea ice hindcast to remove the influence of internal climate variability on the simulated sea ice cover that is present in unconstrained simulations of coupled climate models. We use the Community Earth System Model version 2 (CESM2) as the model, as it includes a sophisticated and widely used sea ice model, CICE5 (Hunke et al., 2015). To produce simulated brightness temperatures from model output, we adapt and use ARC3O, an observational operator that has been developed specifically for sea ice-covered ocean (Burgard et al., 2020a, b). Observation operators (also commonly referred to as satellite simulators, satellite emulators or instrument simulators) offer an opportunity to enable more direct, physically-based model-satellite comparisons. They have been utilized extensively in studying cloud processes in climate models (Bodas-Salcedo et al., 2011) and are becoming more prevalent in the evaluation of other climate model processes (Flato et al., 2013). ARC3O is the first observation operator that has been developed specifically for sea ice from climate models (Burgard et al., 2020a, b). In the following, we will use ARC3O and "satellite simulator" interchangeably.





## 2   Data and Methods

In order to create and evaluate a new metric, namely the simulated earliest snowmelt estimation date, we utilize hindcast
simulations with the ocean-sea ice components from the Community Earth System Model Version 2 (CESM2), the ARC3O
satellite simulator and various satellite data products (DMSP SSM/I-SSMIS brightness temperatures (Meier et al., 2019),
AMSR-E/Aqua brightness temperatures (Cavalieri et al., 2014) and continuous and early melt onset dates based on the DMSP
brightness temperature data (Steele et al., 2019)). All data have been regridded to the same rectilinear grid for consistency.
Detailed results are described for year 2003 to demonstrate the utility of the method.

### 2.1   Satellite brightness temperature data

To evaluate the simulated brightness temperatures we utilize two sets of satellite data, the DMSP SSM/I-SSMIS Daily Polar
Gridded Brightness Temperatures Version 4 (Meier et al., 2019) and the AMSR-E/Aqua Daily L3 Brightness Temperatures
Version 3 (Cavalieri et al., 2014). Both products are gridded to a 25 x 25 km grid and have been regridded to a 0.9 x 1.25°grid
here for comparisons with the model. The DMSP data begin on 9 July 1987 and processing of data is ongoing. The AMSR data
span 1 June 2002 to 4 October 2011. The DMSP data are collected from 19.3, 22.2, 37.0, 85.5 and 91.7 GHz frequency channels
at both horizontal and vertical polarizations. AMSR data are likewise collected at both horizontal and vertical polarizations
from slightly different channels: 6.9, 10.7, 18.7, 23.8, 36.5, and 89.0 GHz.

### 2.2   Melt onset definitions

Early and continuous melt onset dates are taken from the Arctic Sea Ice Seasonal Change and Melt/Freeze Climate Indicators
from Satellite Data Version 1 (Steele et al., 2019), which is based on the DMSP brightness temperatures from 1979-2017. Each
product is derived from a set of weighted parameters based on brightness temperatures at 19.3 GHz and 37.0 GHz (vertically
polarized and hereby referred to as 19.3V and 37V GHz). If no continuous melt onset date is captured by the brightness
temperature parameters, the continuous melt onset date is defined as the last day that ice concentration fall below 80%, and the
data do not provide information on how often this back-up method is employed.

   In previous work that aimed to use seasonal transitions to assess models, several model definitions of the melt onset were
created and compared (Smith and Jahn, 2019). These definitions were based on physical processes associated with melt, but no
single definition was found to directly correspond to the continuous melt onset dates. The two definitions most comparable to
the continuous melt onset (the first based on snowmelt and the second based on surface temperature) are used here to assess if
using simulated brightness temperatures to derive estimations of melt onset provides more direct comparisons between models
and satellite data. As in Smith and Jahn (2019), snowmelt-based melt onset is determined to occur on the first day that snowmelt
is greater than 0.01 cm/day for 5 consecutive days. Surface temperature-based melt onset is determined to occur on the first
day that surface temperature exceeds -1 °C for three consecutive days, as also used in the assessment of melt onset in CMIP6
models (Smith et al., 2020).



## 2.3 The Arctic Ocean Observation Operator (ARC3O)

ARC3O is a newly developed observational operator that simulates Arctic brightness temperatures from two-dimensional model output for the purpose of comparisons with observations (Burgard et al., 2020a, b). The ARC3O brightness temperatures

are simulated using a modified version of the Microwave Emission Model for Layered Snowpacks (MEMLS) extended to sea ice (Tonboe et al., 2006). ARC3O was developed and evaluated to produce brightness temperatures at 6.9 GHz. Additionally, it includes the respective atmospheric corrections for all AMSR frequencies. In this study, brightness temperatures are calculated for ARC3O under "cold conditions", which means that all brightness temperatures are calculated using MEMLS and no assumptions are made about when and where melt is occurring prior to brightness temperature calculation. Note that while

ARC3O was developed and tested for 6.9 GHz, MEMLS has been used in previous studies for higher frequencies than 6.9 GHz (Tonboe, 2010; Tonboe et al., 2011; Willmes et al., 2014; Lee et al., 2017). ARC3O requires atmosphere, ocean and sea ice model output to act as input to the simulator. For the atmosphere, the required modeled variables include 10-meter wind speeds, columnar liquid water, and columnar water vapor. For the ocean and sea ice, the required modeled variables include sea surface temperatures, sea ice concentrations, sea ice thicknesses, surface temperatures at the interface between the atmosphere

and the snow or sea ice, melt pond fraction, snow water equivalent and snow fraction on the sea ice.

In addition to providing input data, ARC3O users can choose whether to use the simply parameterized ARC3O snow and ice profiles or to submit other profiles. Variables that require vertical profiles are layer temperature, layer salinity, layer density, layer thickness, layer wetness, layer correlation lengths, and layer classification into multi-year ice, first-year ice, or snow.

## 2.4 CESM2 JRA-55 ocean-ice hindcast simulation

Natural variations in the climate system inhibit exact replication of the satellite observations in climate models (Kay et al., 2011; Notz, 2015), as observations themselves represent only one of many possible physical outcomes due to the chaotic nature of the climate system (Lorenz, 1963). Here we use a CESM2 (Danabasoglu et al., 2020) ocean-ice hindcast forced by JRA-55 reanalysis-based atmospheric forcing (Kobayashi et al., 2015; Tsujino et al., 2018) to remove the influence of internal climate variability on the simulated sea ice cover, since analysis of large ensemble simulations demonstrates that model definitions of

melt onset can vary by almost a week in their annual pan-Arctic means due to internal variability alone (Smith and Jahn, 2019). The ocean-ice hindcast set-up uses the CESM2 ocean (Smith et al., 2010) and sea ice (Hunke et al., 2015) components at the default CESM2 resolution of 1° nominal resolution, and atmospheric fields from the JRA-55 forcing for ocean-ice models (Kobayashi et al., 2015; Tsujino et al., 2018). By using a hindcast atmosphere instead of the fully coupled version of CESM2, we are able to compare the evolution of simulated brightness temperatures day-by-day without considering internal variability.

We use a spun-up restart from the simulation used by Kim et al. (2020) to run the model for the period 1979 to 2019 with the required output. The spin-up followed the OMIP1 protocol (Griffies et al., 2016), which means 5 cycles of the 1958–2009 forcing. We here analyze year 2003 from the last cycle.





## 2.5 Framework for brightness temperature simulation

As we are interested in snowmelt, and scattering in the snowpack is more prevalent at higher frequencies (Mätzler, 1987; Barber
et al., 1998), we adapted the ARC3O simulator (Burgard et al., 2020a, b) to produce brightness temperatures for 18.7 GHz
instead of 6.9 GHz. The main change this required was an adaptation of the snow wetness parameter in ARC3O, which was set
to zero for 6.9 GHz. To achieve the best agreement with the evolution of the observed brightness temperatures during the melt
season, we added a step-function representation of snow wetness based on the CICE5 daily snowmelt. In the step-function, the
snow wetness is set to 0.2 when snowmelt is less than or equal to 0.2 cm/day (including 0 cm/day), and 0.5 when snowmelt
is greater than 0.2 cm/day. These thresholds were determined through sensitivity testing of the brightness temperature to snow
layer wetness and were found to yield results more comparable to satellite observations than zero snow wetness. The value of
the layer wetness is equal for all snow layers, and held constant at zero for all sea ice layers.

Because the continuous melt onset dates are derived from DMSP brightness temperatures at 19.3V GHz and 37V GHz,
we produce simulated brightness temperatures at 18.7V GHz, the closest AMSR frequency to 19.3V GHz, in order to be as
consistent as possible with the early melt onset and continuous melt onset melt onset products. Evaluation of the 18.7V GHz
AMSR brightness temperatures versus the 19.3V GHz DMSP brightness temperatures show that differences between the two
products are often less than 5 K, and it is therefore appropriate to use the 18.7V GHz frequency for evaluating brightness
temperatures and melt onset dates derived from the satellite simulator. Since the annual cycle of brightness temperatures
is clearly discernible at the 18.7V GHz frequency, we do not create brightness temperatures at 37V GHz, an even higher
frequency than ARC3O was originally designed to simulate, likely requiring further refinements to the representation of snow
on sea ice than were needed for 18.7 GHz.

Other changes to ARC3O are related to the fact that ARC3O was developed to be applied to the largest possible range of
GCMs, regardless of the complexity of their sea ice output (Burgard et al., 2020a, b). So here we leverage the additional model
output from CICE5 to replace the vertical profiles and multi-year ice fractions used in ARC3O with model output.
For the vertical profiles, the layer thickness is derived by equally dividing the snow and ice thicknesses between the respective
number of layers, as in the original ARC3O. CICE5 provides information on three snow layers and eight sea ice layers. Instead
of the default ARC3O simply parameterized profiles, we use temperature and salinity profiles from CICE5 across the sea ice
layers and temperature profiles from CICE5 across the snow layers. Snow salinity is constant at 0 PSU in CICE5. Both the
snow and sea ice profiles are provided for five ice thickness categories. As the differences in the temperature profiles between
the thickness categories are small, we create a weighted mean of the five profiles in order to create profiles representative of
the entire grid cell. For this, the five profiles are weighted by the ice volume in each category at each respective grid cell and
time-step. The three snow layers offer more information about the vertical temperature profile of the snow on sea ice—an
important quantity for detecting melt onset—than the original ARC3O set-up, which used only one snow layer. Based on the
sea ice temperature and salinity profiles derived from CICE5, the density of the sea ice at each layer is calculated following the
original framework of ARC3O, based on Notz et al. (2005). Snow density is held constant at 330 kg/m$^3$, as in CICE5.





In terms of multi-year versus first-year ice, we use the CICE5-provided daily first-year ice fractions based on Lagrangian tracking, instead of the original ARC3O algorithm for determining first or multi-year ice, which defined areas of first-year ice as those where ice thickness reached zero on any day in the past year. Here, sea ice in any grid cell with more than 50% first year ice is considered first year ice, and all other sea ice is considered multi-year ice. The layer correlation length is 0.15 for
snow, 0.25 for first-year ice and 1.5 for multi-year ice. This is a minor simplification from Burgard et al. (2020a, b), in which the correlation length for first-year ice changes to 0.35 below 0.2 m depth.

Feeding daily CESM2 JRA-55 data as input to our slightly adapted version of ARC3O, the evolution and spatial pattern of the brightness temperature in the shelf seas during the spring agrees well with the remotely sensed brightness temperatures (Fig. S1). A high bias tends to correspond to areas of very deep snow (greater than 30 cm) (Fig. S2), as well as areas of multi-
year sea ice (Fig. 2c). In Burgard et al. (2020a, b) a correction of 0.968 was applied to brightness temperatures at all grid cells, and we have modified this approach by applying a correction of 0.92 only to regions of multi-year sea ice to achieve the best agreement with satellite observations (Fig. S1).

## 2.6    New metric: earliest snowmelt estimation date

Using the simulated snowmelt and its signature in the simulated brightness temperatures (Fig. 1), we define a new metric
called the "earliest snowmelt estimation" date. By creating a new metric for snowmelt and applying it in the same way to both observed and simulated brightness temperatures, we are able to reduce the uncertainty related to definition differences between the model and observations. By basing it on a physical process, i.e. the beginning of snowmelt in the model, this metric allows for a process-based, direct model comparison with the remotely-sensed brightness temperatures. This is not possible for the original continuous melt onset from Markus et al. (2009), where the physical processes that it captures are
not exactly clear, and which uses a sophisticated algorithm that weights a combination of three melt onset parameters, using brightness temperatures at two frequencies, as well as employing an ice concentration-based backup method. Nonetheless, we can compare our earliest snowmelt estimation to early and continuous melt onset dates from the algorithm of Markus et al. (2009) to assess their differences and evaluate if different snow and sea ice changes are being captured by the two methods (Sec. 3.2).
The earliest snowmelt estimation is based on the 10-day running mean of the brightness temperatures, so that the estimation is not unduly affected by small day-to-day differences between the AMSR, DMSP and simulated datasets. We then define the earliest snowmelt estimation as the first day between 1 April and 31 June that the brightness temperature crosses a given threshold, which is chosen to reflect when snowmelt begins in the model. The brightness temperatures on the first day of simulated snowmelt vary spatially (Fig. 1). In order to keep the definitions based solely on brightness temperatures so that
consistent comparisons between model and satellite data are possible, we set two brightness temperature thresholds. The best agreement is found with the following thresholds: if the brightness temperature at a given grid cell is greater than 242 K on 1 January, the brightness temperature has to be greater than 259 K to be counted as earliest snowmelt date, and if the brightness temperature is less than or equal to 242 K on 1 January, the brightness temperature has to be greater than 239 K (Fig. 2). In the model the boundary between these thresholds aligns with boundary between multi-year and first year ice. Hence, this approach



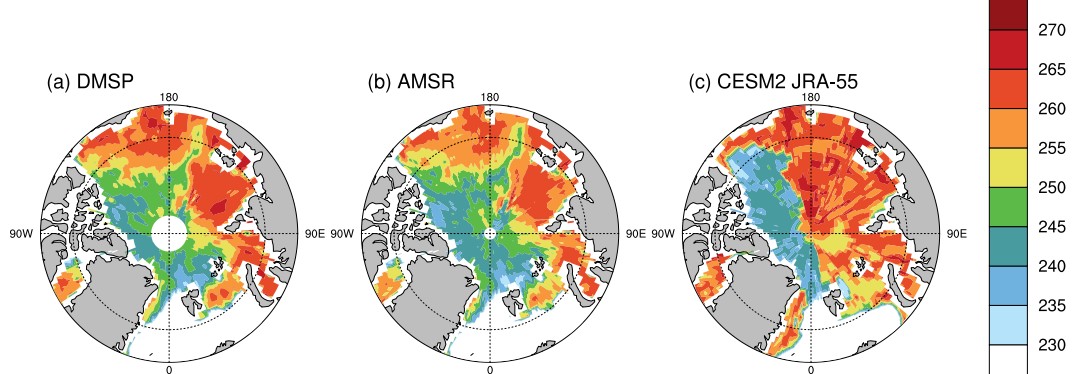

**Figure 1.** The brightness temperature (K) on the first day in 2003 at each grid point that CESM2 JRA-55 snowmelt is greater than 0.01 cm/day in (a) DMSP (b) AMSR and (c) CESM JRA-55.

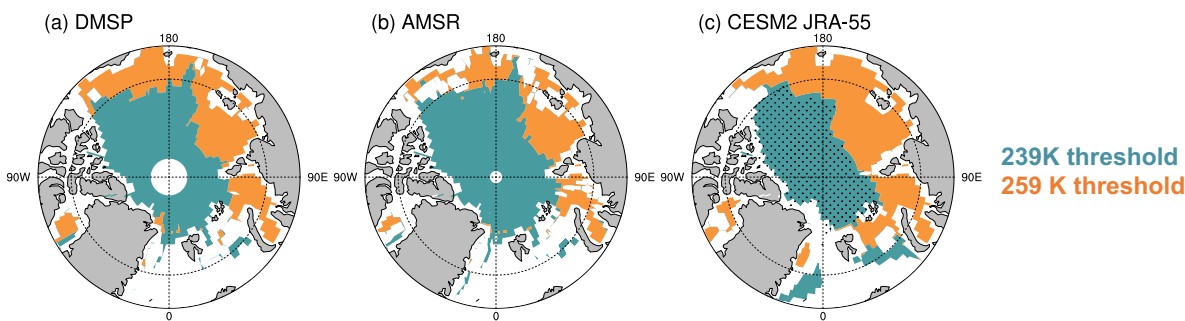

**Figure 2.** Areas where the earliest snowmelt estimation employs a 259 K threshold (orange) and a 239 K threshold (turquoise) in 2003 for (a) DMSP (b) AMSR and (c) CESM JRA-55. In panel (c) the area of multi-year ice from CESM JRA-55 is stippled.

is very similar to that used by Markus et al. (2009), where the brightness temperature on 1 April is used to determine whether the sea ice is multi-year or first year ice, and where different thresholds at the 19.3V GHz frequency are used to determine one of the melt onset parameters.

## 3    Results

The results are presented in three sections, beginning with the evaluation of the simulated brightness temperatures in Section
3.1. We then compare the new metric, the earliest snowmelt estimation dates, to various representations of melt onset in Section 3.2, demonstrating the utility of the new metric for improving model-satellite data comparisons and bias detection. Finally, we provide an assessment of how much snowmelt occurs prior to the detection of melt onset using the early and continuous melt onset algorithms in Section 3.3.



### 3.1 Simulated brightness temperatures at the beginning of the melt season

Spring brightness temperatures produced at 18.7V GHz by the satellite simulator are comparable to those from the DMSP and AMSR satellite data in terms of their variability, magnitudes and, most importantly, temporal evolution (Fig. 3). First, the simulated brightness temperatures, which are produced based on CESM2 hindcast simulation output, capture the day-to-day variability seen in the satellite data (Fig. 3), consistently matching local maxima and minima. The variability in the observed brightness temperatures is large, with single-day changes at times exceeding +/-10 K, and the simulated brightness

temperatures generally capture such changes well. Furthermore, the magnitudes of the brightness temperatures are consistent with observations in large areas of the Arctic and for much of the year (Figs. 3, S1). In the marginal seas, the simulator consistently produces brightness temperatures of similar magnitude to satellite data from January through June (Fig. 3a-c). Simulated brightness temperatures tend to deviate from the satellite observations when the ice concentration declines, showing less agreement in their variability (Fig. 3a-c). In the central Arctic, the variability is also well captured in the model, but the

magnitude of the brightness temperatures is higher than observed early in the year. Once the sea ice concentration declines, the magnitude of the brightness temperature in the central Arctic in the model becomes lower than observed and the variability no longer matches the observations (Fig. 3d). However, this does not affect evaluation of the earliest snowmelt estimation, which occurs earlier in the spring, before substantial sea ice loss. Over the course of the late winter and spring, brightness temperatures show a small decline or are steady from January to April in both the simulator and observations, followed by a

clear increase from April until May, when snowmelt increases (Fig. 4) and simulated ice concentrations first start to decrease (Fig. 3). Since the simulated brightness temperatures tend to correctly capture the evolution of brightness temperatures from January through June, they are suitable for assessing the beginning of the melt season.

### 3.2 Utilizing earliest snowmelt estimation dates

#### 3.2.1 Physical insights into the impact of melt processes on brightness temperature

Simulated brightness temperatures are valuable for understanding the physical processes associated with sea ice seasonality because they are directly comparable to other model variables that are relevant to brightness temperature-sensitive processes, such as snowmelt and ice concentration changes (Smith, 1998; Markus et al., 2009). Here we have used the dependence of the brightness temperature on snowmelt to define the snowmelt estimation threshold when the earliest snowmelt in the model occurs (Section 2.6). Since the temporal evolution and variability of the simulated brightness temperatures agree well with

satellite data, especially in the spring (Sect. 3.1), agreement between the simulated and observed brightness temperatures in terms of the earliest snowmelt estimation suggests that snowmelt occurs at about that time in reality as well. Furthermore, for periods when simulated and observed brightness temperatures agree, we can use the model to provide insights into which processes are captured by existing melt onset definitions.

The earliest snowmelt estimation based on brightness temperature thresholds successfully captures the beginning of snowmelt

on sea ice in the model at individual locations, even as the timing in snowmelt varies across regions of the Arctic (Fig. 4). Furthermore, the earliest snowmelt estimation agrees well between the simulation and DMSP and AMSR satellite data across the



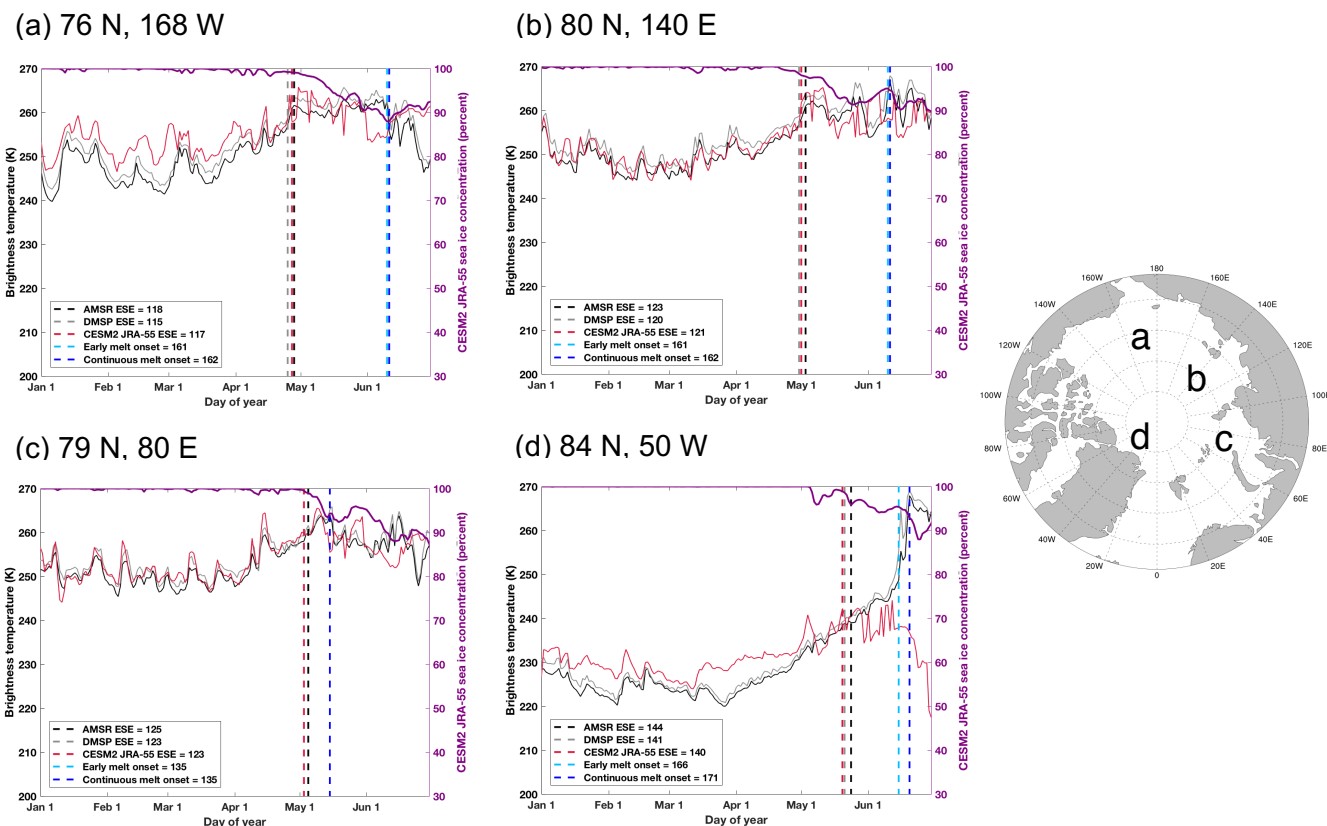

**Figure 3.** AMSR (black), DMSP (grey) and CESM2 JRA-55 (red) daily brightness temperatures in 2003 at approximately (a) 76 N, 168 W (b) 80 N, 140 E (c) 80 N, 80 E and (d) 84 N, 50 W. In each panel, daily CESM2 JRA-55 ice concentration (purple) is shown in percent at the respective location. Dashed lines show the AMSR (black), DMSP (grey) and CESM2 JRA-55 (red) earliest snowmelt estimation dates (ESE) as well as the early melt onset date (light blue) and the continuous melt onset date (dark blue).

Arctic (Fig. 3). At this time, ice concentration gradually begins to decline as well, but at many locations still exceeds 95% (Fig. 3). Hence, the increase in the brightness temperature that triggers the earliest snowmelt estimation suggests that this metric is capturing the start of snowmelt in both the model and reality, as captured by the satellite data.

Both early and continuous melt onset based on the DMSP brightness temperatures fall later in the season than the earliest snowmelt estimation. In addition, both the early and continuous melt onset tend to occur after approximately 10% of the ice in the grid cell has been lost (Fig. 3). The continuous melt onset date tends to fall near the middle of the seasonal cycle of snowmelt on the sea ice (Fig. 4). Early melt onset is less consistent as to when in the seasonal cycle of snowmelt it occurs, in some places occurring on the same day as continuous melt onset (Fig. 4c), and in other places about a week earlier (Fig. 4d).

While the melt onset algorithms are designed to capture snow and ice melt (Markus et al., 2009), it has not previously been possible to analyze if one or both of the two processes are occurring at the time of diagnosed melt onset across the Arctic. Based





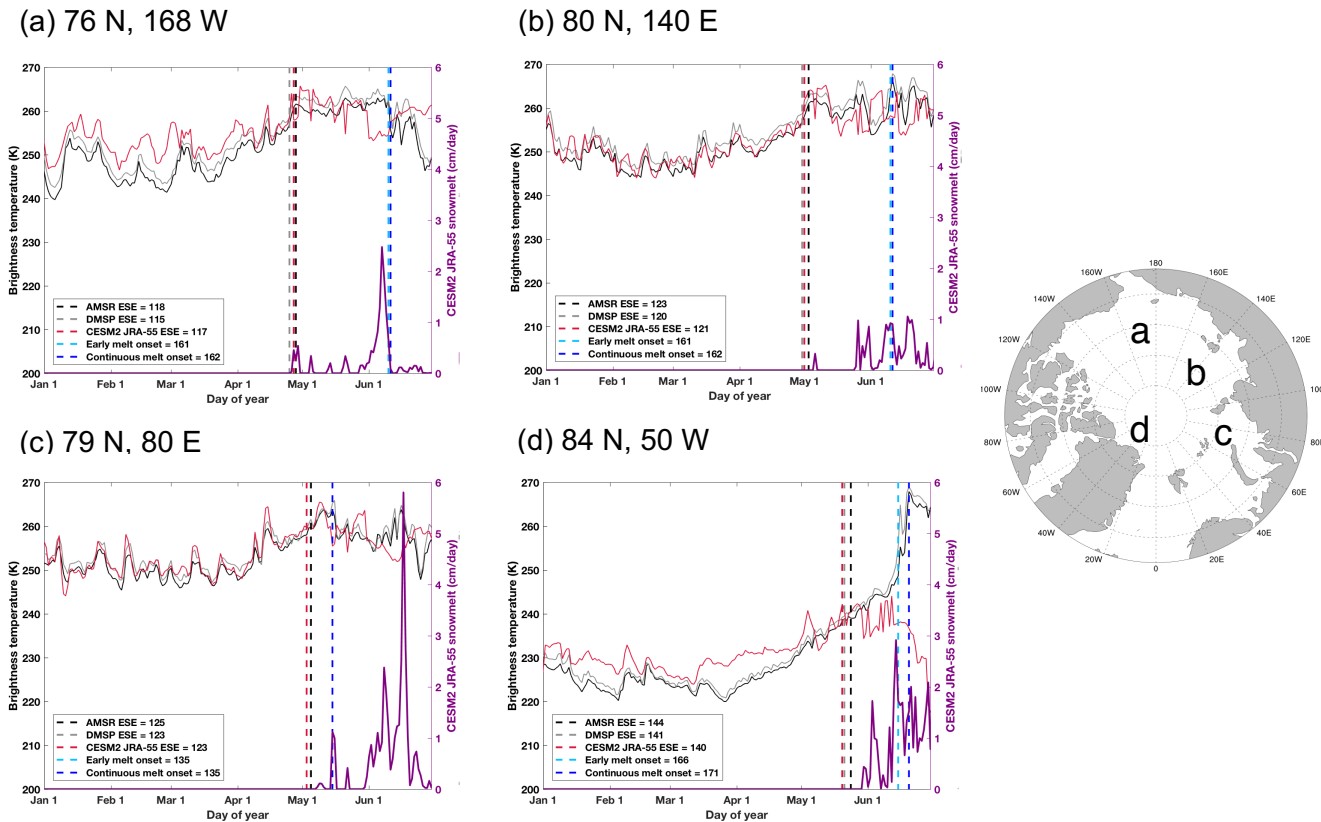

**Figure 4.** AMSR (black), DMSP (grey) and CESM2 JRA-55 (red) daily brightness temperatures in 2003 at approximately (a) 76 N, 168 W (b) 80 N, 140 E (c) 80 N, 80 E and (d) 84 N, 50 W. In each panel, daily CESM2 JRA-55 snowmelt (purple) is shown in cm/day at the respective location. Dashed lines show the AMSR (black), DMSP (grey) and CESM2 JRA-55 (red) earliest snowmelt estimation dates as well as the early melt onset date (light blue) and the continuous melt onset date (dark blue).

on what we have found, it is clear that the earliest snowmelt estimation captures a different physical change in the snow and sea ice than the early and continuous melt onset dates: the earliest snowmelt estimation occurs at the beginning of snowmelt, while early and continuous melt onset occur after substantial seasonal snow and ice loss has occurred.

**3.2.2 Diagnosing model biases based on direct comparisons**

The earliest snowmelt estimation provides a method of determining melt that can be applied to both observed and simulated brightness temperatures. This not only allows us to leverage other model variables to inform our process-understanding of snow and sea ice (Sec. 3.2.1), but also provides more direct comparisons between climate models and satellite observations. The earliest snowmelt estimation of the simulated brightness temperatures from the ocean-ice CESM2 hindcast shows a similar

spatial pattern to the earliest snowmelt estimations from the DMSP and AMSR satellite data (Fig. 5). Simulated earliest





snowmelt estimation dates occur latest in areas north of the Canadian Arctic Archipelago, the Laptev Sea and in the northern Kara Sea. Direct comparison between the model and observations shows that the simulated earliest snowmelt dates, while generally occurring at a similar time to the DMSP and AMSR earliest snowmelt estimations, tend to occur earlier than those derived from satellite data (Fig. 6). Modeled earliest snowmelt dates fall slightly late in the Beaufort Sea, as well as the northern

areas of the Laptev, Kara and Barents Sea (Fig. 6). Since the regions of late bias fall along the line differentiating thresholds in the calculation of earliest snowmelt dates (Sec. 2.2), these biases may be related to differences in the position of multi-year ice between the model and observations. The largest differences between the model and observations is in the Central Arctic, where simulated earliest snowmelt dates fall over 45 days earlier in some areas. By comparing the same quantity (brightness temperature), processed in the same way (earliest snowmelt estimation), we can clearly discern that the earliest

snowmelt estimation in the ocean-ice CESM2 hindcast is occurring too early in the Central Arctic (Fig. 6). Finally, AMSR and DMSP satellite products disagree on the sign of the bias in the southern region of the Laptev Sea, thus a model bias cannot be confidently diagnosed in this region.

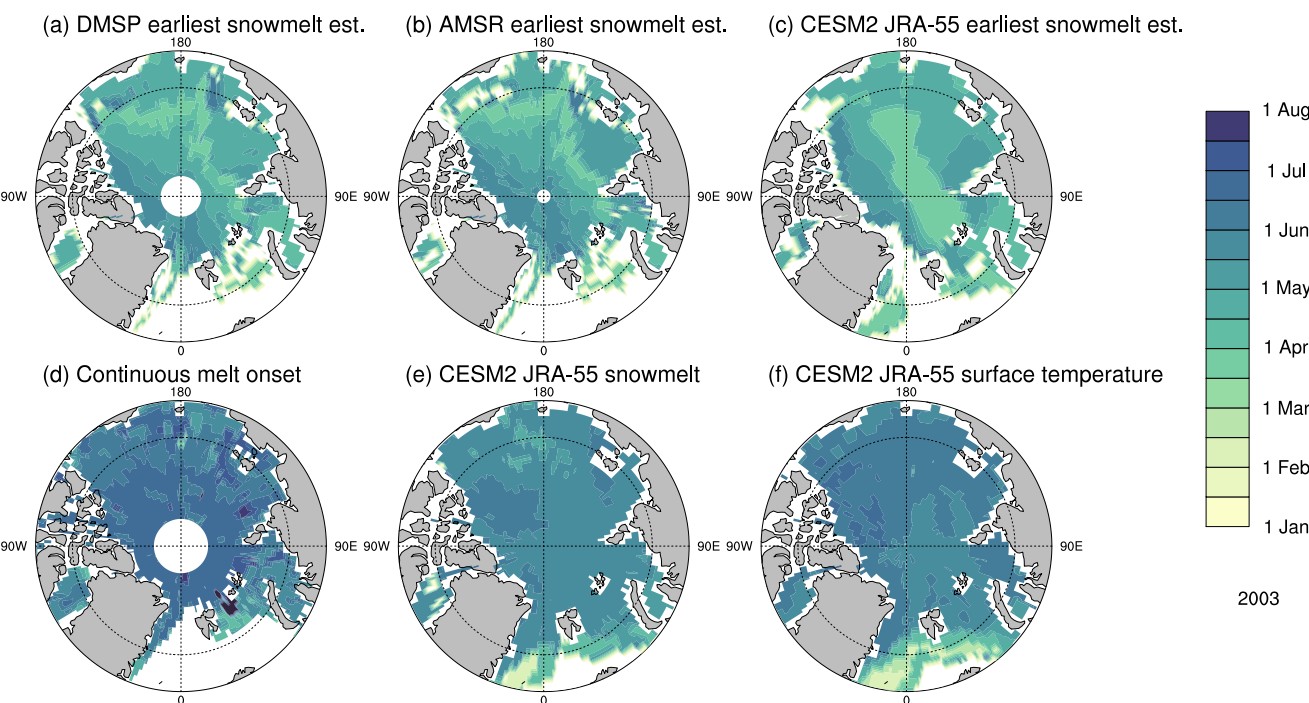

**Figure 5.** Comparison of the (a) DMSP earliest snowmelt estimation dates (b) AMSR earliest snowmelt estimation dates (c) CESM2 JRA-55 earliest snowmelt estimation dates (d) continuous melt onset dates (based on DMSP) (e) CESM2 JRA-55 snowmelt melt onset dates and (f) CESM2 JRA-55 surface temperature melt onset dates at each grid cell, for 2003.

The direct comparison improves upon previous model-satellite data comparisons involving the continuous melt onset. Because brightness temperatures were not previously simulated using the CESM, other definitions based on known physical

**Figure 6.** Differences between melt onset dates. The CESM2 JRA-55 earliest snowmelt estimation dates minus (a) DMSP earliest snowmelt estimation dates and (b) AMSR earliest snowmelt estimations dates; (c) CESM2 JRA-55 snowmelt melt onset dates minus continuous melt onset dates (based on DMSP) (d) CESM2 JRA-55 surface temperature melt onset dates minus continuous melt onset dates (based on DMSP). All for 2003.

processes in the model were previously used to compare melt onset dates between climate models and satellite observations (Smith and Jahn, 2019; Smith et al., 2020). These process-based definitions were developed based on criteria that represent melt (snowmelt and a change in surface temperature, see Section 2.2), and the specifications of each definition have been





refined to best characterize continuous melt onset, though they do so without any known basis for which physical processes are most relevant. These model definitions of melt onset fall about 10-40 days later than the earliest snowmelt estimation dates (Fig. 7). The snowmelt and surface temperature melt onset dates fall earlier than the early and continuous melt onset dates from DMSP, and tend to fall closer to one of these two metrics than to the earliest snowmelt estimation dates (Fig. 7). While snowmelt and surface temperature changes sometimes occur close to the melt onset dates based on the algorithms by Markus et al. (2009) (Fig. 7a,b), they don't always (Fig. 7c,d), and the physical reasons for when they agree and when they do not agree are not clear. Hence, this comparison is less process-specific than the newly possible comparisons between the earliest snowmelt estimations from the simulated brightness temperatures.

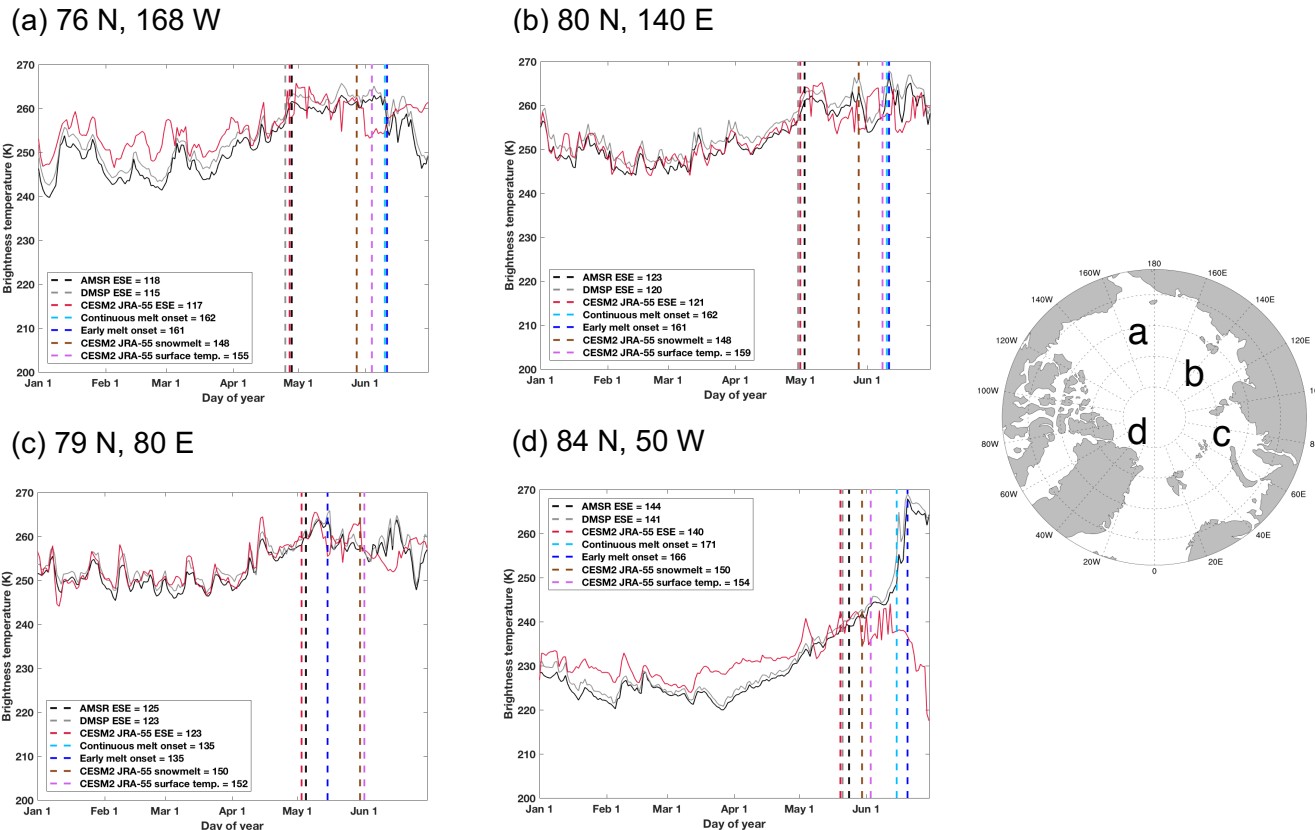

**Figure 7.** AMSR (black), DMSP (grey) and CESM2 JRA-55 (red) daily brightness temperatures in 2003 at approximately (a) 76 N, 168 W (b) 80 N, 140 E (c) 80 N, 80 E and (d) 84 N, 50 W. Dashed lines show the AMSR (black), DMSP (grey) and CESM2 JRA-55 (red) earliest snowmelt estimation dates (ESE) as well as the early melt onset date (light blue), the continuous melt onset date (dark blue), the CESM2 JRA-55 snowmelt melt onset date (brown) and the the CESM2 JRA-55 surface temperature melt onset date (pink).

Because the snowmelt and surface temperature melt onset dates generally fall earlier than the observed continuous melt onset dates (Figs. 5, 6) and the simulated earliest snowmelt estimation falls earlier than the AMSR and DMSP earliest snowmelt





estimations in most regions of the Arctic, the direction of the model bias has generally not changed in implementing a new
comparison. This suggests that the underlying biases persist throughout the melt season. This bias detection adds value to our
assessment of model representations of the melt, because the processes captured are known and the quantities being compared
between the model and observations are identical, reducing the uncertainty from differences in definitions.

### 3.3   Quantifying the snowmelt detected by melt onset algorithms

Direct comparisons between climate models and satellite observations also provide opportunities for the models to inform us
about the physical processes captured by satellite data. In Section 3.2.1, we show that substantial snowmelt occurs in the model
before the melt onset algorithms detect early or continuous melt. Since the evolution in the simulated brightness temperatures
agrees well with the satellite brightness temperatures during the melt season, in particular in the marginal seas, we can use the
model to provide some insights into how much snowmelt occurs by the time melt onset is detected by the satellite algorithm.
In order to quantify the amount of snowmelt before diagnosed melt onset, we calculate the cumulative snowmelt starting on
the day of maximum snow depth and ending on the the dates of early and continuous melt onset. Because total snowmelt is
dependent on snow depth, we normalize the cumulative snowmelt by the maximum snow depth at each grid cell. This yields
a unitless quantity referred to here as the normalized snowmelt, which estimates the amount of snowmelt that occurs prior to
early and continuous melt onset at each grid cell (Fig. 8). The normalized snowmelt ranges between 0 and 1 across most of the
Arctic. However this value can and does exceed 1 in some locations, particularly in the inflow regions where late-spring storms
may be expected, since snow accumulation can still occur after the date of maximum snow depth.

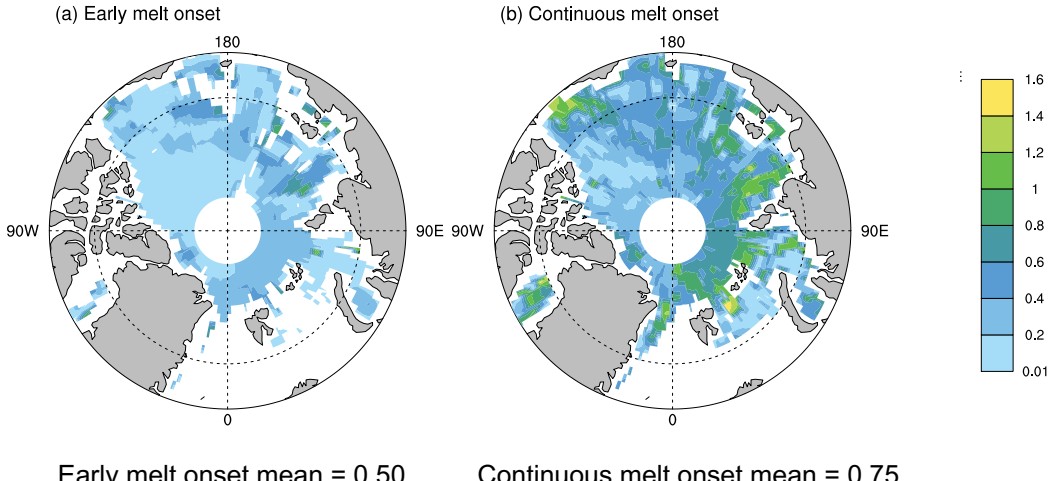

**Figure 8.** Normalized snowmelt in 2003 prior to the (a) early melt onset date and (b) continuous melt onset date at each grid cell. Spatial
mean values are listed below each panel.





Because the normalized snowmelt falls below 1 across large areas of the Arctic and the spatial variability is not large in either the case of early or continuous melt onset (Fig. 8), we are able to take spatial means to provide pan-Arctic estimations of the normalized snowmelt. The mean normalized snowmelt prior to early melt onset is about 0.5, while the the mean normalized snowmelt prior to continuous melt onset is about 0.75. This means that approximately half of the snow melts in the model before the detection of early melt onset by the satellite algorithm and approximately three quarters melts before continuous melt onset

detection. And while models can have biases, they can still provide valuable insights such as these that would be impossible to obtain without a large number of spatially-distributed in situ snowmelt observations. Normalized snowmelt therefore serves as an example of the novel, process-oriented analysis that can be done through more direct comparisons between models and satellite observations in the Arctic.

## 4    Conclusions

We find that brightness temperatures produced at 18.7 GHz using a ocean-ice CESM2 hindcast and the ARC3O satellite simulator agree well with observed brightness temperatures in our period of interest, from January to the end of June. This combination is therefore a useful tool for investigating Arctic sea ice and snow melt processes from a novel perspective. In the marginal seas, simulated brightness temperatures are similar to AMSR and DMSP brightness temperatures in terms of their magnitude and variability in the spring months, which are of interest for evaluating melt (Fig. 3).

Since the variability of the simulated brightness temperatures agree well with satellite data, we use the model to develop a new metric for melt onset, called the earliest snowmelt estimation. The earliest snowmelt estimation is designed to capture the beginning of snowmelt, based on when snowmelt first occurs in the simulation from the brightness temperatures (Fig. 4), and can hence be applied consistently to both the model and the satellite data. We find that the earliest snowmelt estimation dates generally agree well between the model and AMSR and DMSP, but generally occur about 5-30 days earlier in the model

compared to AMSR and DMSP in most regions of the Arctic (Figs. 5,7). The early model bias in the earliest snowmelt date is also seen in the comparison of the satellite-based continuous melt onset dates (Markus et al., 2009; Steele et al., 2019) with other CESM2 melt onset definitions derived from the snowmelt and surface temperature variables that are designed to approximate the melt onset as captured by the satellite data (Figs. 5,7). By taking advantage of the more direct model-to-observation comparisons enabled by the simulator, we are able to show that this persistent early occurrence of melt for different

metrics in CESM2 JRA-55 is indeed a model bias, and not due to definition differences. Regions showing a late bias in CESM2 JRA-55 occur along the threshold line of the earliest snowmelt estimation definition and are likely affected by uncertainty in the location of multi-year sea ice.

    In addition to assessing the model performance by using the satellite simulator, we are also able to provide insights into the physical processes captured by satellite data in a novel way. We find that compared to the PMW satellite-algorithm based early

and continuous melt onset dates (Markus et al., 2009; Steele et al., 2019), the earliest snowmelt date occurs much earlier in the year. This means that rather than at the start of the melt, the early and continuous melt onset dates occur after substantial loss of simulated snow and sea ice (Figs. 4, 3). Using normalized snowmelt analysis in the model, we find that approximately one half





of the snow melts before early melt onset is detected and approximately three quarters melts by the time continuous melt onset is detected. While these values may be affected by model biases, they provide useful quantitative insights that were previously

impossible to quantify. Hence, normalized snowmelt analysis demonstrates the power of direct, process-oriented comparisons between models and satellite observations.

To conclude, using a satellite simulator that provides brightness temperatures from model output allows us to perform the most direct assessment of simulated onset of snowmelt against satellite observations to date, as well as to provide physical insights into the meaning of various melt onset-related dates. By doing so, we overcame several previous limitations on model-

satellite comparisons, opening the door for more robust and physically relevant model assessments in the future. To enable such comparisons in future climate simulations, daily temperature and salinity profile information as well as daily snowmelt and first-year ice fractions should be saved. However, even without this output needed to run ARC3O, simulated daily snowmelt can be used to compare models against the earliest snowmelt estimation dates from satellite brightness temperatures.

*Code and data availability.* Documentation and installation guidelines for ARC3O are publicly available at https://arc3o.readthedocs.io/en/

latest/. Code adapted and created for this work will be archived with a DOI while the paper is under review and can be found in its pre-publication version at the following link: https://drive.google.com/drive/folders/1sY6_Jh5Y6Lw2omvKtmhLYblrsIdZO8gn?usp=sharing. AMSR and DMSP data are publicly available at https://nsidc.org/. Earliest snowmelt estimation dates will be published through the Arctic Data Center with a DOI while the paper is under review.

*Acknowledgements.* We thank Who Kim (NCAR) for providing us with restarts for the CESM2 hindcast simulations and David Bailey

who provided expertise and advise on CICE5 and the CESM2. We also acknowledge helpful discussions about the idea of a melt onset simulator with Andrew Roberts and Sinead Farrell. A. Smith's contribution is supported by the Future Investigators in NASA Earth and Space Science Grant no. 80NSSC19K1324 and the National Science Foundation Graduate Research Fellowship grant no. DGE 1144083. A Jahn's acknowledges funding by NSF-OPP CAREER grant 1847398. For the CESM2 hindcast simulations, we would like to acknowledge high-performance computing support from Cheyenne (doi:10.5065/D6RX99HX) provided by NCAR's Computational and Information Systems

Laboratory, sponsored by the National Science Foundation. NCAR is sponsored and the CESM project is primarily supported by the National Science Foundation. We thank all the scientists, software engineers, and administrators who contributed to the development of CESM2.



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
