# Peer review of "Improving model-satellite comparisons of sea ice melt onset with a satellite simulator"

_The Cryosphere, 2021_

## Author Comment (AC1)

**Response to RC1**

The paper uses a brightness temperature satellite simulator to diagnose the timing of melt onset in model simulations, and compare this new metric to other metrics of melt onset. Overall, the paper is well written and worthy of publication subject to the minor corrections listed below:

We thank the referee very much for their positive assessment and their very constructive comments that will help us improve the manuscript. Responses to the minor corrections are below.

1. Line 49: Write out the ACR3O acronym here as this is its first use.
Thank you for noticing this, the acronym stands for "Arctic Ocean Observation Operator" and will be written out at first use.

2. Line 61: What method of regridding was used?
The nearest neighbor source-to-destination regridding method was used and this will be noted in the text.

3. Line 73: "Each .." Describe the method in slightly more detail - How does brightness temperature change as melt occurs?
Text will be added to this section to provide more detail on the Markus et al., 2009 methodology (used in the Steele et al., 2019 dataset).

4. Line 85: "Surface temperature" - clarify whether this is ice, snow or either surface temperature.
The surface temperature-based definition is based on the CESM model variable surface temperature, and is the sea ice temperature if no snow is present and the snow temperature if snow is present. This will be noted in the text.

5. Line 123: What is the justification for the step function as opposed to another functional form?
In the absence of an empirical function to describe snow wetness, we chose to prioritize the simplicity of the functional form in order to reduce the uncertainty introduced by this estimation. While other functional forms were not evaluated, we will include a brief addition to the text describing the sensitivity of the results to the step function thresholds.

6. Line 138: "So here" -> "Therefore" or another replacement.
We will replace "So here" with "Therefore".

7. Line 141: "CICE provides..." I believe CICE can provide the number of layers specified so clarify this isn't the only choice possible. Maybe something like: "CICE was configured to provide"

We will make this change in the text as suggested.

8. Lines 140-150: Clarify what thermodynamic model CICE is used? Mushy layer? Bitz-Lipscomb 1999?

Here CICE5 was configured to use the default thermodynamic scheme within the CESM, which is the mushy-layer thermodynamics. This will be noted in the text.

9. Line 152: Lagrangian tracking - I am unfamiliar with this functionality in CICE - is it documented/referenced somewhere? There is a FY ice tracer, but it is not lagragian tracked, instead advected with the Eulerian transport scheme.

Thank you for this clarification. The phrase "Lagrangian tracking" does not apply, as we are utilizing the FY ice output variable. This will be corrected in the text.

10. Line 154: Clarify what the correlation length is.

The correlation length is a metric of the scatterer size. Although it is a variable that is quite well understood and quantifiable for snow (Mätzler, 2002; Proksch et al., 2015; Lemmetyinen et al., 2018), its quantification in sea ice is not clear and its values not well known. We therefore use values based on past experiments conducted by Rasmus T. Tonboe (Burgard et al., 2020a) and this will be noted in the text.

11. Line 159: "(greater than 30cm)(Fig. S2)" -> "(greater than 30cm; Fig. S2)"

We will correct this in the text.

12. Line 180-185: The two thresholds are presumably justified from a bimodal nature of the brightness temperatures. It would be useful to see histograms demonstrating this bimodal nature.

Thank you for this suggestion. We will create histograms of the brightness temperatures and assess if they can be well-integrated in the manuscript's Supplement and discussion of results.

13. Line 185: "with boundary" -> "with the boundary"

We will correct this in the text.

14. Figures 1, 5, 6, 8: Add the field name and units to the colorbars.

Field names and units will be added to the colorbars.

15. Figures 3, 4, 7: Add markers to the map of the Arctic ocean showing exactly where the a,b,c,d points are.

Latitude and longitude markers will be added to the maps.

16. References: "SIMIP community ... 2020": This doesn't appear to be how to cite: https://agupubs.onlinelibrary.wiley.com/action/showCitFormats?doi=10.1029%2F2019GL086749

Thank you for raising this point. The citation of the SIMIP paper was changed at Wiley after the initial publication for purely administrative reasons related to the funding of the publishing charges being carried centrally by the Max-Planck-Society in Germany. As this change was implemented for purely administrative reasons, rather than reflecting the intended, agreed authorship of the paper, we would prefer to keep the reference as reading like the original authorship "SIMIP community, 2020" if journal guidelines allow.

---

## Author Comment (AC2)

**Response to RC2**

This paper describes a novel use of model simulated brightness temperatures to compute a new metric to identify the timing of earliest snowmelt on Arctic sea ice. The paper compares the simulated brightness temperatures and earliest snowmelt dates with brightness temperatures and a melt onset data set from satellite data.

The paper is well written, and the study is clearly presented. I agree with comments from the prior reviewer and recommend that this paper is published pending a few additional clarifications as described in my comments below.

We thank the referee very much for their positive assessment and their constructive comments that will help us improve the manuscript. Responses to the comments are below.

1. L61: Why was 2003 selected as the sample year? Is it representative of a normal (non- anomalous) melt onset year? Or something else?
We selected the year 2003 to incorporate both DMSP and AMSR-E brightness temperatures, as AMSR-E begins in June 2002, and to avoid the anomalously low sea ice extent years of 2007 and 2012. This decision will be noted in the text and a figure showing a time series of annual average melt onset dates will be added to the Supplement for reference.

2. L73: Why is the Steele et al. 2019 dataset used instead of the Markus melt onset product directly? Are they not the same data?
It is correct that the melt onset dates from Steele et al., 2019 are the same as Markus et al., 2009. The Steele et al., 2019 dataset, which provides additional variables based on ice concentration, is used here for consistency and ease of comparison with past work on spring sea ice transition metrics (Smith et al., 2020). We will clarify this in the text.

3. L206-207: To what extent do you think error in the observations or in the simulated brightness temperature contribute to the divergence between the simulated and observed brightness temperatures in the central Arctic (i.e, Figure 3d) seen after the SIC declines? What might the physical reason for this big difference be?
Differences between the simulated and observed brightness temperatures after the SIC declines are likely related to differences in SIC between the model and observations at each grid cell. If the SIC declines faster in the model than observed, the brightness temperature will also decrease more quickly with the presence of more open water. Our focus on earlier melt processes reduces the impacts of these differences on our analysis. We will clarify this in the text.

4. L288: Please add proper names for the geographic locations that are considered "inflow regions".

We will add proper names for the geographic locations.

---

## Author Response (AR1)

**Response to RC1**

The paper uses a brightness temperature satellite simulator to diagnose the timing of melt onset in model simulations, and compare this new metric to other metrics of melt onset. Overall, the paper is well written and worthy of publication subject to the minor corrections listed below:

We thank the referee very much for their positive assessment and their very constructive comments that have helped us improve the manuscript. A point-by-point reply detailing the revisions is provided below. New text that has been added to the manuscript is in **bold and italics**.

1. Line 49: Write out the ACR3O acronym here as this is its first use.
"Arctic Ocean Observation Operator" (ARC3O) is now written out at first use in the abstract and main text.

2. Line 61: What method of regridding was used?
That you for noticing that information was missing. The text was altered as follows:
"All data have been regridded to the same rectilinear grid for consistency ***using the nearest neighbor source-to-destination method.***"

3. Line 73: "Each .." Describe the method in slightly more detail - How does brightness temperature change as melt occurs?
The text was altered as follows:
"Early and continuous melt onset dates are derived from a set of weighted parameters based on brightness temperatures at 19.3 GHz and 37.0 GHz (vertically polarized and hereby referred to as 19.3V and 37V GHz). ***These parameters are designed to capture changes in the brightness temperature driven by increasing emissivity associated with increasing snow and ice wetness.***"

4. Line 85: "Surface temperature" - clarify whether this is ice, snow or either surface temperature.
It is the surface temperature of snow or ice. To clarify, the text now reads:
"***Melt onset derived from the surface temperature of either the sea ice or overlying snow*** is determined to occur on the first day that surface temperature exceeds -1 degree C for three consecutive days, as also used in the assessment of melt onset in CMIP6 models (Smith et al., 2020)."

5. Line 123: What is the justification for the step function as opposed to another functional form?

Thank you for asking this. To clarify this, the text now reads:
"In the step-function, the snow wetness is set to 0.2 when snowmelt is less than or equal to 0.2 cm/day (including 0 cm/day), and 0.5 when snowmelt is greater than 0.2 cm/day. These thresholds were determined through sensitivity testing of the brightness temperature to snow layer wetness and were found to yield results more comparable to satellite observations than zero snow wetness. *Simulated brightness temperatures are slightly more sensitive to changes in the step function threshold than the assigned wetness value, but sensitivity analysis showed that varying the step function resulted in brightness temperature changes of less than 10 K.* The value of the layer wetness is equal for all snow layers, and held constant at zero for all sea ice layers."

6. Line 138: "So here" -> "Therefore" or another replacement.
As suggested, we replaced "So here" with "Therefore".

7. Line 141: "CICE provides..." I believe CICE can provide the number of layers specified so clarify this isn't the only choice possible. Maybe something like: "CICE was configured to provide"
8. Lines 140-150: Clarify what thermodynamic model CICE is used? Mushy layer? Bitz-Lipscomb 1999?
To address points 7 and 8 the text was altered as follows:
"*Here, CICE5 was configured using the mushy-layer thermodynamics scheme and to provide information* on three snow layers and eight sea ice layers."

9. Line 152: Lagrangian tracking - I am unfamiliar with this functionality in CICE - is it documented/referenced somewhere? There is a FY ice tracer, but it is not lagragian tracked, instead advected with the Eulerian transport scheme.
The reviewer is correct, it was not Lagrangian tracking. The text now reads as follows:
"In terms of multi-year versus first-year ice, we use the CICE5-provided daily first-year ice fractions based on *an Eulerian transport scheme*, instead of the original ARC3O algorithm for determining first or multi-year ice, which defined areas of first-year ice as those where ice thickness reached zero on any day in the past year."

10. Line 154: Clarify what the correlation length is.
To clarify this, the text now reads as follows:
"The layer correlation length, *a metric of the scatterer size*, is 0.15 for snow, 0.25 for first-year ice and 1.5 for multi-year ice. This is a minor simplification from Burgard et al., 2020a,b), in which the correlation length for first-year ice changes to 0.35 below 0.2 m depth. *As explained in Burgard et al. (2020a), the correlation length is a variable that is well understood and quantifiable for snow (Mätzler, 2002; Proksch et al.,*

*2015; Lemmetyinen et al., 2018), but not necessarily for sea ice. The values used are therefore based on past experiments conducted by Rasmus T. Tonboe (Burgard et al., 2020a).*

11. Line 159: "(greater than 30cm)(Fig. S2)" -> "(greater than 30cm; Fig. S2)"
Thank you, this has been corrected in the text as suggested.

12. Line 180-185: The two thresholds are presumably justified from a bimodal nature of the brightness temperatures. It would be useful to see histograms demonstrating this bimodal nature.
Thank you for this suggestion. We have created the suggested histograms (for May 1 2003) and included them as a Supplemental figure in the revised manuscript, with new text added to refer to them. "In order to keep the definitions based solely on brightness temperatures so that consistent comparisons between model and satellite data are possible, we set two brightness temperature thresholds *based on the brightness temperature distributions, which are skewed high in all products and appear more bimodal in the CESM2 (Fig. S4).*"

[Figure]

**12. continued**

The suggestion from reviewer 1 to create histograms of the brightness temperatures to justify the thresholds picked prompted us to also create histograms of the melt onset dates, out of curiosity. In the histograms, it is more clearly visible that CESM JRA-55 shows a higher proportion of earliest melt estimation dates falling on 1 April (the first day possible by our methods) than in the satellite observations. In the original submission, we started checking for earliest melt estimation dates on 1 April to avoid transient winter melt events and for similarity to Markus et al., 2009, which uses this date for a metric differentiating first-year and multi-year ice. However, as shown below, beginning the check on 1 March reduces the number of earliest melt estimation dates falling by default on the first possible day. It also more clearly demonstrates the early bias detected in CESM JRA-55. Thus we have altered our approach to begin checking on 1 March instead of 1 April. Figures 5 and 6 have been recreated reflecting this change, but the key results are unchanged. Locations selected for Figures 3, 4, and 7 are unaffected.

The text has been altered as follows:
Section 2.6
"We then define the earliest snowmelt estimation as the first day between 1 March and 31 June that the brightness temperature crosses a given threshold, which is chosen to reflect when snowmelt begins in the model. ***We begin on 1 March to avoid transient winter melt events and to reduce the number of earliest melt estimation dates falling by default on the first possible day in the CESM JRA-55 relative to later start dates.***"

Section 2.6
Hence, this approach is  similar to that used by Markus et al., 2009, where the brightness temperature on 1 April is used to determine whether the sea ice is multi-year or first year ice, and where different thresholds at the 19.3V GHz frequency are used to determine one of the melt onset parameters.

Section 3.2.2
"The largest differences between the model and observations is in the Central Arctic, where simulated earliest snowmelt dates fall over ***60 days*** earlier in some areas."

Conclusions
We find that the earliest snowmelt estimation dates generally agree well between the model and AMSR and DMSP, but generally occur about 5-***45*** days earlier in the model compared to AMSR and DMSP in most regions of the Arctic (Figs. 5-7).

[Figure]

Distributions of earliest melt estimation dates when beginning on 1 April 2003.

[Figure]

Distributions of earliest melt estimation dates when beginning on 1 March 2003.

[Figure]

(a) DMSP earliest snowmelt est.     (b) AMSR earliest snowmelt est.     (c) CESM2 JRA-55 earliest snowmelt est.

(d) Continuous melt onset     (e) CESM2 JRA-55 snowmelt     (f) CESM2 JRA-55 surface temperature

1 Aug
1 Jul
1 Jun
1 May
1 Apr
1 Mar
1 Feb
1 Jan

2003

Updated Fig. 5

(a) CESM2 JRA-55 minus DMSP     (b) CESM2 JRA-55 minus AMSR

(c) CESM2 JRA-55 snowmelt minus CMO     (d) CESM2 JRA-55 surface temp. minus CMO

-1
-5
-15
-30
-45
-60
-75

days

Updated Fig. 6

13. Line 185: "with boundary" -> "with the boundary"
Thank you, this has been corrected in the text.

14. Figures 1, 5, 6, 8: Add the field name and units to the colorbars.
As suggested, field names and units have been added to the colorbars.

15. Figures 3, 4, 7: Add markers to the map of the Arctic ocean showing exactly where the a,b,c,d points are.
As suggested, latitude and longitude markers have been added to the maps.

16. References: "SIMIP community ... 2020": This doesn't appear to be how to cite: https://agupubs.onlinelibrary.wiley.com/action/showCitFormats?doi=10.1029%2F2019GL086749
The citation of the SIMIP paper was changed at Wiley after the initial publication for purely administrative reasons related to the funding of the publishing charges being carried centrally by the Max-Planck-Society in Germany. As this change was implemented for purely administrative reasons, rather than reflecting the intended, agreed authorship of the paper, we would prefer to keep the reference as reading like the original authorship "SIMIP community, 2020" if journal guidelines allow.

**Response to RC2**

This paper describes a novel use of model simulated brightness temperatures to compute a new metric to identify the timing of earliest snowmelt on Arctic sea ice. The paper compares the simulated brightness temperatures and earliest snowmelt dates with brightness temperatures and a melt onset data set from satellite data.

The paper is well written, and the study is clearly presented. I agree with comments from the prior reviewer and recommend that this paper is published pending a few additional clarifications as described in my comments below.

We thank the referee very much for their positive assessment and their constructive comments that have helped us improve the manuscript. A point-by-point reply detailing the revisions is provided below. New text that has been added to the manuscript is in **bold and italics**.

1. L61: Why was 2003 selected as the sample year? Is it representative of a normal (non- anomalous) melt onset year? Or something else?
The text was altered as follows and a figure showing a time series of pan-Arctic average melt onset dates has been added to the Supplement, with 2003 circled in red:
"Detailed results are described for year 2003 to demonstrate the utility of the method. ***The year 2003 was chosen for multiple reasons: it is not an extreme year in terms of pan-Arctic average melt onset (Fig. S1), it falls after the start of the AMSR-E data (June 2002) and is not an anomalously low sea ice extent year (such as 2007 or 2012).***"

Additionally, text was added to the Conclusions:
"By taking advantage of the more direct model-to-observation comparisons enabled by the simulator, we are able to show that this persistent early occurrence of melt for different metrics in CESM2 JRA-55 is indeed a model bias, and not due to definition differences. Regions showing a late bias in CESM2 JRA-55 occur along the threshold line of the earliest snowmelt estimation definition and are likely affected by uncertainty in the location of multi-year sea ice. ***This assessment was conducted for 2003, which is not an extreme year in terms of pan-Arctic average melt onset, and earliest snowmelt estimation dates from other years should be evaluated in the future.***"

[Figure]

2. L73: Why is the Steele et al. 2019 dataset used instead of the Markus melt onset product directly? Are they not the same data?

Yes, the melt onset data are the same in the two datasets. To reflect that, the text now reads

"Early and continuous melt onset dates are taken from the Arctic Sea Ice Seasonal Change and Melt/Freeze Climate Indicators from Satellite Data Version 1 (Steele et al., 2019), which is based on the DMSP brightness temperatures from 1979-2017. *The Steele et al., 2019 dataset provides melt onset dates calculated using the method established by Markus et al., 2009 as well as additional variables based on ice concentration. It is used here for consistency and ease of comparison with past work on spring sea ice transition metrics (Smith et al., 2020).*"

3. L206-207: To what extent do you think error in the observations or in the simulated brightness temperature contribute to the divergence between the simulated and observed brightness temperatures in the central Arctic (i.e, Figure 3d) seen after the SIC declines? What might the physical reason for this big difference be?

Thanks for asking. To address this, the text now reads as follows:

Once the sea ice concentration declines, the magnitude of the brightness temperature in the central Arctic in the model becomes lower than observed and the variability no longer matches the observations (Fig. 3d), *likely due to differences in sea ice concentration between the model and observations at individual grid cells*.

4. L288: Please add proper names for the geographic locations that are considered "inflow regions".

As requested, we added the proper names, and the text now reads as follows: "However this value can and does exceed 1 in some locations, particularly in **_the Barents and Chukchi Seas_** where late-spring storms may be expected, since snow accumulation can still occur after the date of maximum snow depth."